# Genome-wide screening identified *SEC61A1* as an essential factor for mycolactone-dependent apoptosis in human premonocytic THP-1 cells

Akira Kawashima[1], Mitsuo Kiriya[1], Junichiro En[1,2], Kazunari Tanigawa[3], Yasuhiro Nakamura[4], Yoko Fujiwara[1], Yuqian Luo[1,5], Keiji Maruyama[4], Shigekazu Watanabe[4], Masamichi Goto[6], Koichi Suzuki[1]*

**1** Department of Clinical Laboratory Science, Faculty of Medical Technology, Teikyo University, Tokyo, Japan, **2** Department of Occupational Therapy, School of Health Science, International University of Health and Welfare, Narita, Japan, **3** Department of Molecular Pharmaceutics, Faculty of Pharma-Science, Teikyo University, Tokyo, Japan, **4** Center for Promotion of Pharmaceutical Education & Research, Faculty of Pharma-Science, Teikyo University, Tokyo, Japan, **5** Department of Laboratory Medicine, Nanjing Drum Tower Hospital and Jiangsu Key Laboratory for Molecular Medicine, Nanjing University Medical School, Nanjing, China, **6** Department of Pathology, Kagoshima University Graduate School of Medical and Dental Sciences, Kagoshima, Japan

* koichis0923@med.teikyo-u.ac.jp

## Abstract

Buruli ulcer is a chronic skin disease caused by a toxic lipid mycolactone produced by *Mycobacterium ulcerans*, which induces local skin tissue destruction and analgesia. However, the cytotoxicity pathway induced by mycolactone remains largely unknown. Here we investigated the mycolactone-induced cell death pathway by screening host factors using a genome-scale lenti-CRISPR mutagenesis assay in human premonocytic THP-1 cells. As a result, 884 genes were identified as candidates causing mycolactone-induced cell death, among which *SEC61A1*, the α-subunit of the Sec61 translocon complex, was the highest scoring. CRISPR/Cas9 genome editing of *SEC61A1* in THP-1 cells suppressed mycolactone-induced endoplasmic reticulum stress, especially eIF2α phosphorylation, and caspase-dependent apoptosis. Although previous studies have reported that mycolactone targets SEC61A1 based on mutation screening and structural analysis in several cell lines, we have reconfirmed that SEC61A1 is a mycolactone target by genome-wide screening in THP-1 cells. These results shed light on the cytotoxicity of mycolactone and suggest that the inhibition of mycolactone activity or SEC61A1 downstream cascades will be a novel therapeutic modality to eliminate the harmful effects of mycolactone in addition to the 8-week antibiotic regimen of rifampicin and clarithromycin.

## Author summary

Buruli ulcer is a chronic skin disease caused by the bacterium *Mycobacterium ulcerans*. The disease mainly affects children in West Africa, and the skin ulcers are induced by

**Data Availability Statement:** All relevant data are within the manuscript and its Supporting Information files.

**Funding:** This study was supported by grants from the Japan Society for the Promotion of Science (JSPS) KAKENHI (grant numbers JP16K09808 and JP19K07557 (AK and MG)), Takeda Science Foundation (AK), GSK Japan Research Grant 2020 (AK) and AMED (grant number 18jm0510004h0001 (KS)). The funders had no role in study design, data collection and analysis, decision to publish, or preparation of the manuscript.

**Competing interests:** The authors have declared that no competing interests exist.

mycolactone, a toxin produced by the bacteria. The mycolactone diffuses through the skin, killing cells, creating irreversible ulceration, and weakening host immune defenses. However, the cytotoxic pathway induced by mycolactone remains largely unknown. We evaluated the mycolactone-induced cell death pathway by screening host factors using a genome-scale knockout assay in human premonocytic THP-1 cells. We identified 884 genes that are potentially involved in mycolactone-induced cell death, of which SEC61A1, the α-subunit of the Sec61 translocon complex, was the highest ranking. Knockout of *SEC61A1* in THP-1 cells resulted in suppression of endoplasmic reticulum stress and caspase-dependent apoptosis induced by mycolactone. These results suggest that SEC61A1 is an essential mediator of mycolactone-induced cell death.

## Introduction

Buruli ulcer (BU) is a chronic skin disease caused by *Mycobacterium ulcerans*, a pathogen that lives in aquatic environments. The disease is reported mainly in West Africa, but cases are also found in parts of Asia, South America, the Western Pacific and Australia [1]. Children are most affected in West Africa and, if left untreated, they can develop lifelong disabilities and disfigurements, often causing stigma. Thus, the World Health Organization designated BU as a neglected tropical disease.

The macrolide exotoxin mycolactone is the virulence factor responsible for BU induces skin destruction, chronic skin ulceration and osteomyelitis in severe cases [2]. Mycolactone is detected at high levels in skin lesions; however, it also diffuses at the systemic level and suppresses immune responses [3]. An animal study showed that mycolactone is responsible for tissue damage and immune suppression [4]. Intradermal inoculation of *M. ulcerans* in Guinea pigs produced skin ulcerations similar to those of human BU [4]. We previously showed that *M. ulcerans* clones lacking a plasmid that encodes genes essential for mycolactone synthesis had no pathologic effects in mice [5]. Intravenously injected mycolactone accumulates in the spleen of mice, and the mycolactone has a higher affinity for mononuclear cell subsets than for neutrophils [6]. Indeed, it was suggested that mycolactone affects circulating lymphocytes and alters their protective immune function, which contributes to the immune evasion of *M. ulcerans* [6]. The essential host factors and signal transduction pathways that mediate the action of mycolactone are an area of active investigation.

*In vitro* studies showed that mycolactone blocks the translocation of nascent proteins across the ER membrane, and similar inhibitors such as CT7, CT8, and ipomoeassin F directly target the ER Sec61 translocon [7–12]. The SEC61α amino acid mutation R66G partly rescued mycolactone-induced cell death and protein translocation in HEK293 cells [11]. A structural study also showed that mycolactone binds the Sec61 translocon [13]. On the other hand, a haploid genetic screen revealed that the histone methyltransferase SETD1B is a mediator of mycolactone-induced cell death in KBM-7 cells, a chronic myelogenous leukemia cell line [14]. The activating transcription factor 4 (ATF4) also affects cell death pathways through the ER stress response in HeLa cells [15]. In this paper, we employed genome-wide screening with CRISPR/Cas9 technology to further the analysis of molecular pathways in mycolactone-induced cell death.

Recent advancements in CRISPR/Cas9 technology have enabled gene disruption studies in mammalian cells on a genome-wide scale [16]. A genome-wide CRISPR knockout (GeCKO) screening strategy has been utilized to investigate pathogen–host interactions [17] and the cytotoxic pathways induced by toxic reagents [18]. Human lenti-CRISPR/Cas9 knockout

pooled libraries (v2) containing six small guide (sg) RNAs per gene, target the conserved 5′ coding exons of 19,050 human genes. Compared with siRNAs, the CRISPR/Cas9 system has higher targeting specificity and fewer off-target effects [19]. Therefore, we employed CRISPR knockout screening with human GeCKO sgRNA libraries to conduct comprehensive and unbiased loss-of-function screens to identify genes necessary for mycolactone-induced cell death in human premonocytic THP-1 cells.

## Methods

### Cell cultures and reagents

Human premonocytic THP-1 cells were obtained from the American Type Culture Collection (Manassas, VA, USA). Cells were cultured in RPMI 1640 medium (Sigma-Aldrich, St. Louis, MO, USA) supplemented with 10% fetal calf serum (Sigma-Aldrich) and 100 U/mL penicillin/ 100 μg/mL streptomycin (Sigma-Aldrich). Lenti-X 293T cells (Takara Bio Inc., Otsu, Shiga, Japan) were cultured in Dulbecco's modified Eagle's medium (Wako Pure Chemical Industries, Ltd., Osaka, Japan) supplemented with 10% fetal calf serum, 100 U/mL penicillin and 100 μg/mL streptomycin. The cell number and viability were measured by the trypan blue exclusion assay using an automatic cell counter (Countess; Life Technologies, Carlsbad, CA, USA).

Synthetic mycolactone A/B was provided as an ethanol-diluted solution (1 mg/mL) by Dr. Yoshito Kishi, Department of Chemistry and Chemical Biology, Harvard University [20]. The mycolactone A/B solution was diluted in the culture medium at final concentrations of 3, 30 and 300 ng/mL. Ethanol diluted in the culture medium was used as the solvent control. Actinomycin D (ActD; Sigma-Aldrich) was used to induce apoptosis as a control. Thapsigargin (Wako) was used to induce ER stress as a control. z-Asp-Glu-Val-Asp-fluoromethyl ketone (Z-DEVD-FMK), a pan-caspase inhibitor, was purchased from Cayman Chemical (Ann Arbor, MI, USA). The DEVD-FMK peptide inhibits caspase-3,7, 8 and 9 [21, 22].

### Lentivirus production and screening for mycolactone resistance

The human GeCKO v2 pooled library was a gift from Feng Zhang (Addgene, Cambridge, MA, USA; #1000000048). The GeCKO library was divided into two libraries, A and B. Each library was co-transfected with pCMV-VSV-G (RIKEN, Wako, Saitama, Japan) and psPAX2 (gift from Didier Trono; Addgene plasmid #12260; http://n2t.net/addgene:12260; RRID: Addgene_12260) into Lenti-X 293T cells using PEI-MAX (Polysciences, Warrington, PA, USA). Lentiviral particles from the culture supernatant were filtered through a 0.45-μm ultralow protein-binding filter (Merck Millipore, Bedford, MA, USA) and concentrated by ultracentrifugation ($50,000 \times g$ for 2 h). For lentivirus transduction, THP-1 cells ($2.0 \times 10^8$) were incubated with lentiviral vectors for 16 h at a multiplicity of infection of 0.3 in the presence of 8 μg/mL polybrene (Sigma-Aldrich). Cells were plated in a T225 flask (Thermo Fisher Scientific, Waltham, MA, USA) in RPMI 1640 medium containing 8 μg/mL polybrene for 3 days, followed by incubation with 0.5 μg/mL puromycin (Sigma-Aldrich) for at least 1 week. THP-1 cells harboring GeCKO libraries A and B were treated with 30 ng/mL mycolactone for 1 week. Surviving cells were reseeded, expanded to $3.0 \times 10^7$ cells, and harvested for genomic DNA sequencing.

### Genomic DNA sequencing and analysis

Genomic DNA was purified from $3.0 \times 10^7$ THP-1 cells using the QIAamp DNA Mini Kit (Qiagen, Valencia, CA, USA) and subjected to PCR to amplify lenti-CRISPR v2 sgRNAs using

the following primers: (sense) 5′-TCTTGTGGAAAGGACGAAC-3′ and (antisense) 5′-TAGGCA CCGGATCAATTGC-3′. Adaptors were added to each end of the PCR products for next-generation sequencing using the following PCR primers: (sense, read 1 sequencing primer) 5′-TCGT CGGCAGCGTCAGATGTGTATAAGAGACAGTCTTGTGGAAAGGACGAAACACC-3′ and (antisense, read 2 sequencing primer) 5′- GTCTCGTGGGCTCGGAGATGTGTATAAGAGACAGTA CACGACATCACTTTCCC-3′. The products from the second round of PCR were subjected to electrophoresis and extracted from agarose gels using the MinElute Gel Extraction Kit (Qiagen) according to the manufacturer's instructions. The PCR products were sequenced using the next-generation sequencer NovaSeq 6000 (Illumina, San Diego, CA, USA).

sgRNAs enriched in the mycolactone-resistant population were ranked using Model-based Analysis of Genome-wide CRISPR-Cas9 Knockout (MAGeCK) (v0.5.8) [23], in which the read counts of each sequence from mycolactone-selected cells were compared with matched read counts from unselected control cells. MAGeCK consists of read count normalization, sgRNA ranking and gene ranking. Briefly, read counts from different samples were first median normalized to adjust for the effect of library size and read count distribution. MAGeCK was run using the default parameters, with the human GeCKO v2 pooled library (Addgene) used as a reference. MAGeCK ranks each sgRNA based on p-values calculated from a negative binomial model, which is used to test whether sgRNA abundance differs significantly from the mean and variance of all samples. All sgRNAs targeting each gene were then ranked and summarized into one score for the gene (gene score) using a modified robust ranking aggregation (RRA) algorithm [24]. The RRA algorithm uses a probabilistic model for aggregation that is robust to noise and that facilitates the calculation of significance probabilities for all elements in the final ranking.

## Secondary assessment of CRISPR screening hits by cell viability assay

Each of the top-ranking sgRNAs from the GeCKO screening was inserted into the Cas9-encoding lenti-CRISPR v2 vector (52961; Addgene) [16], and the vector was transfected into Lenti-X 293T cells using PEI-MAX (Polysciences) together with three packaging plasmids (encoding gag-pol, rev and vesicular stomatitis virus glycoprotein). The lentiviral particles from the culture supernatant were concentrated by ultracentrifugation (50,000 × $g$ for 2 h). THP-1 cells were incubated with lentiviral vectors for 16 h in the presence of 8 μg/mL polybrene (Sigma-Aldrich) for lentivirus transduction, then selected with 0.5 μg/mL puromycin (Sigma-Aldrich) for at least 1 week. Surviving cells were reseeded in a 10 cm dish and expanded to use for the experiments. Cells were also transduced with sgRNA targeting the enhanced green fluorescent protein (*EGFP*) gene as a control [25]. After selection, $2.0 \times 10^5$ cells were seeded in each well of a 24-well plate in 500 μL RPMI 1640 medium containing 5% fetal bovine serum. The following day, mycolactone was added at a final concentration of 30 ng/mL, and the cells were incubated for 48, 96 and 144 h before assessing cell viability by the trypan blue exclusion assay. Five replicates per condition were performed in each of three independent experiments.

## Total RNA isolation and real-time PCR

Total RNA was purified using the RNeasy Plus Mini Kit (Qiagen), and cDNA was synthesized using the High-Capacity cDNA Reverse Transcription Kit (Applied Biosystems, Waltham, MA, USA) as descried previously [26, 27]. Real-time PCR was performed using the Thermal Cycler Dice Real Time System III and Fast SYBR Green Master Mix (both from Applied Biosystems) according to the manufacturer's instructions as described previously [28, 29]. The sgRNA sequences are listed in S1 Table. Real-time PCR analysis was conducted in triplicate.

The resulting mRNA levels were normalized to *GAPDH* mRNA levels and expressed relative to the control mRNA levels. The PCR primers are listed in S2 Table.

## Western blot analysis

Cells were lysed in buffer containing 150 mM NaCl, 1% Nonidet P-40, 0.5% sodium deoxycholate, 0.1% SDS and 50 mM Tris pH 8.0 for 1 h (4˚C). The supernatant was collected after centrifugation, and 10 μg protein was used for Western blot analysis, as describe previously [28, 30, 31]. Briefly, the proteins were separated on NuPage 4–12% Bis-Tris gels (Invitrogen, Waltham, MA, USA) by electrophoresis and transferred to nitrocellulose membranes using i-Blot gel transfer stacks (Invitrogen). The membrane was washed with PBS containing 0.1% Tween 20 (PBST) and incubated in blocking buffer (PBST containing 5% nonfat milk) for 1 h. Then, the membrane was incubated with primary antibodies (all at 1:1000 dilution) at 4˚C overnight. The primary antibodies used were rabbit anti-caspase-3 (Cell Signaling Technology #9662, Beverly, MA, USA), rabbit anti-cleaved caspase-3 (Asp175) (Cell Signaling Technology #9661), rabbit anti-SEC61A1 (D4K2Z) (Cell Signaling Technology #14867), rabbit anti-eukaryotic initiation factor (eIF2)α (D7D3) (Cell Signaling Technology #5325), rabbit anti-phospho-eIF2α (Ser51) (D9D8) (Cell Signaling Technology #3398), rabbit anti- protein kinase-like ER kinase (PERK) (C33E10) (Cell Signaling Technology #3192), rabbit anti-phospho-PERK (Thr980) (16F8) (Cell Signaling Technology #3179), rabbit anti-activating transcription factor (ATF6) (D4Z8V) (Cell Signaling Technology #65880), rabbit anti-inositol requiring enzyme-1 (IRE1)α (14C10) (Cell Signaling Technology #3294) and mouse anti-β-actin (Sigma-Aldrich #5441). After washing with PBST, the membranes were incubated with horseradish peroxidase-labeled goat anti-rabbit IgG (Cell Signaling Technology #7074) or goat anti-mouse IgG (Cell Signaling Technology #7076) as secondary antibodies (both at 1:1000 dilution). The horseradish peroxidase signal was detected using Immunostar LD reagent (Wako Pure Chemical), and chemiluminescence was analyzed using the C-DiGit blot scanner (LI-COR, Lincoln, NE, USA). Original images of Western blots are shown in S4 Fig.

## Genome mutation analysis

To confirm the SEC61A1 genome mutation induced by the SEC61A1-targeting sgRNA, we extracted genomic DNA from *EGFP*-knockout and *SEC61A1*-knockout THP-1 cells using the QIAamp DNA Mini Kit (Qiagen) and subjected the genomic DNA to PCR to amplify the *SEC61A1* sgRNA targeting site using the following primers: (sense) 5′- GCCTGGCGTTGAAT TGGTG-3′ and (antisense) 5′- AAGTGTGAGGGGCTACTCAA-3′. PCR was performed using the Thermal Cycler Dice (Takara Bio, Shiga, Japan). Briefly, the PCR mixture was first denatured for 5 min at 94˚C, followed by 15 cycles of three-temperature PCR consisting of denaturation for 30 sec at 94˚C, annealing for 30 sec at 65˚C and extension at 72˚C for 45 sec. The PCR products were analyzed by 2% agarose gel electrophoresis. The SEC61A1 targeting site was also analyzed by direct sequencing with (sense) 5′- GCCTGGCGTTGAATTGGTG using the Fasmac sequencing service (Fasmac, Atsugi, Japan).

## Measurement of caspase activity

Activation of caspase-dependent apoptosis activity was analyzed using the fluorochrome-labeled inhibitors of caspases (FLICA) Caspase-3/7 Assay Kit (Immunochemistry Technologies, Bloomington, MN, USA) according to the manufacturer's instructions [22]. Briefly, THP-1 cells plated on 8-well cover glass chambers (IWAKI, Chuo, Tokyo, Japan) were incubated with mycolactone (30 or 300 ng/mL) or ActD (1 μM) for 24 h. Cells were then labeled with Z-DEVD-FMK, a cell-permeable fluorogenic substrate used to monitor activated

caspase-3/7, for 24 h and washed with apoptosis wash buffer. Nuclei were counter stained with Hoechst 33342, and fluorescence was detected using confocal laser-scanning microscopy (FV-10i; Olympus, Shinjuku, Tokyo, Japan). More than 1,000 cells were counted to evaluate the percentage of FLICA-positive cells.

## Statistical analysis

Data are expressed as the mean ± SEM. Unpaired t tests were used to assess differences between two groups. A p-value $<0.05$ was considered to represent statistical significance.

## Results

### Synthetic mycolactone-induced cell death in THP-1 cells via apoptosis

To assess the effect of mycolactone, the major virulence factor of *M. ulcerans*, on tissue macrophages (histiocytes), we treated human premonocytic THP-1 cells with mycolactone and assessed cell death. The concentrations of mycolactone used in this study were determined by mass spectrometry analysis of skin and serum samples from patients with BU, as reported previously [3, 32]. Significant cell death was induced by 30 ng/mL synthetic mycolactone at 48 h after treatment in THP-1 cells, as evidenced by the trypan blue exclusion assay (Fig 1A). This concentration is 6 to 8 times higher than the previously reported concentrations used in other cell types [32, 33]. Under the same conditions, Hoechst 33342 staining revealed condensed, fragmented nuclei with much brighter fluorescence, indicating apoptotic DNA fragmentation (Fig 1B). Treatment of THP-1 cells with the pan-caspase inhibitor Z-DEVD-FMK before exposure to mycolactone resulted in significant suppression of mycolactone-induced cell death (Fig 1C), suggesting that the cell death induced by mycolactone was attributed mostly to apoptosis rather than cytotoxicity (necrosis). Western blot analysis showed that mycolactone treatment activated caspase-3, based on the presence of cleaved caspase-3 (Fig 1D, molecular weight: 19,000 and 17,000). The appearance of cleaved caspase-3 was completely blocked by the pan-caspase inhibitor Z-DEVD-FMK (Fig 1D). Taken together, these data suggest that mycolactone induced caspase-3-dependent apoptosis in THP-1 cells.

### Genome-wide CRISPR/Cas9-mediated screening identified SEC61A1 as a factor responsible for mycolactone-induced cell death

To further investigate the underlying molecular mechanism of mycolactone-induced cell death, we performed GeCKO screening in THP-1 cells (Fig 2A). Cells were genome-edited using the lentivirus-based human GeCKO v2 library consisting of 123,411 sgRNAs targeting 19,050 protein-coding genes and 1,000 control genes. Transfection was performed at a low multiplicity of infection ($<0.3$) to ensure that the cells received no more than one sgRNA. Cells were then treated with 30 ng/ml mycolactone for 1 week, and the surviving cells were recovered and reseeded to use for genomic DNA sequencing. The copy number of each sgRNA was determined by deep sequencing using the next-generation sequencer Novaseq 6000. The MAGeCK tool was used to analyze the deep sequencing data, and the RRA algorithm was used to compute gene-level scores, as described in the Methods. The screening results were visualized using volcano plots (Fig 2B). The x-axis of the volcano plots indicates enriched sgRNA counts, and the y-axis indicates significant p-values from MAGeCK RRA comparisons of vehicle versus mycolactone-treated cells. As a result, 884 candidate genes mediating mycolactone-induced cell death were identified (p $<0.05$) (Fig 2B and S1 Data).

To verify the potential host factors required for mycolactone-induced cell death, the top 10 genes with the highest RRA values were selected (S1 Fig and Table 1) and further validated.

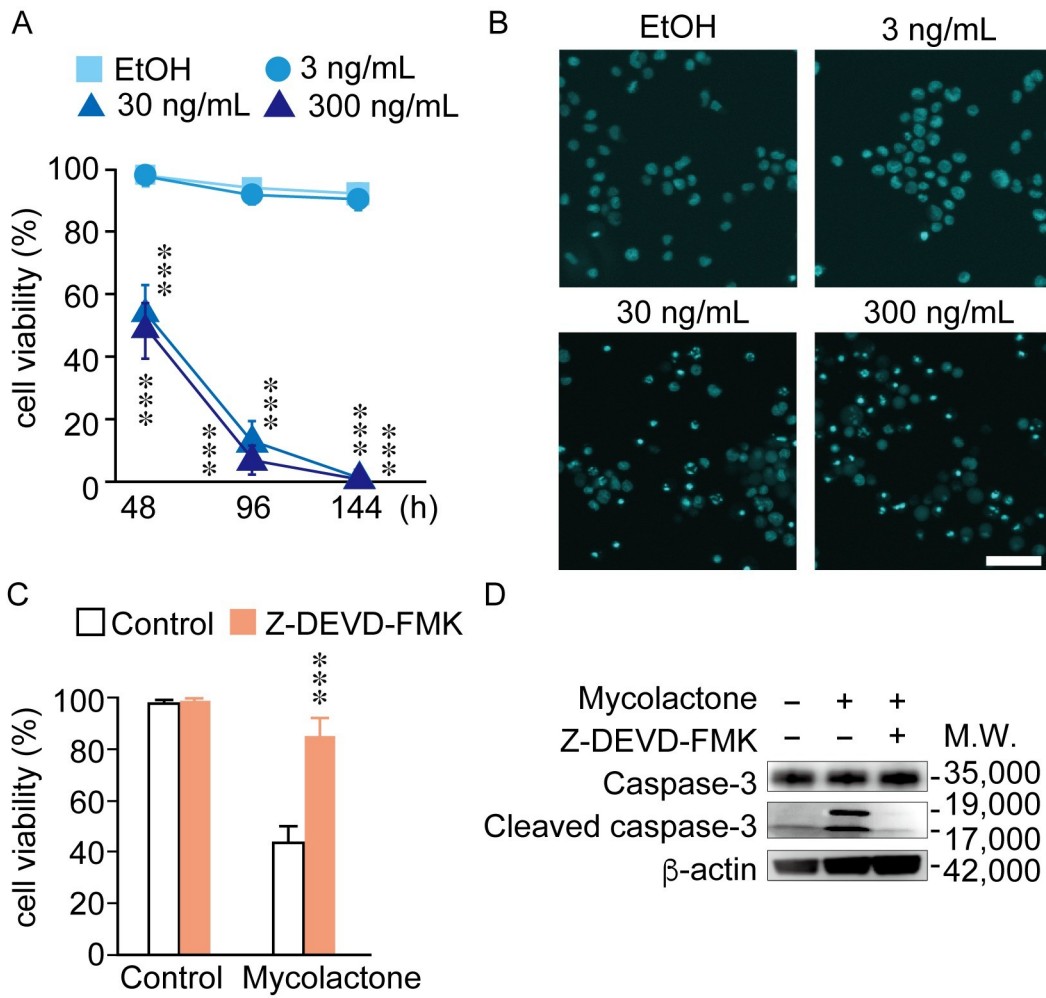

**Fig 1. Synthetic mycolactone-induced caspase-3-dependent apoptosis in THP-1 cells.** (A) Viability of THP-1 cells treated with mycolactone at a final concentration of 3, 30 or 300 ng/mL for 48, 96 or 144 h. Cells were harvested, and viability was determined by the trypan blue exclusion assay. ***: p <0.005 (n = 5) compared with the viability of ethanol (EtOH)-treated cells. (B) Fluorescence images of THP-1 cells treated with or without mycolactone at 3, 30 or 300 ng/mL for 48 h, followed by Hoechst 33342 nuclear staining. Fluorescence was observed using the FV10i confocal laser scanning microscope. Scale bar: 50 μm. (C) Viability of THP-1 cells treated with mycolactone (30 ng/mL, 48 h) in the presence or absence of the pan-caspase inhibitor Z-DEVD-FMK (20 μM, 24 h). Cell viability was determined by the trypan blue exclusion assay (n = 5). ***: p < 0.005. (D) Western blot analysis of caspase-3 and cleaved caspase-3 in lysates from THP-1 cells treated with 30 ng/mL mycolactone for 48 h in the presence or absence of Z-DEVD-FMK (20 μM). β-actin was used as a loading control.

Thus, THP-1 cells were transduced with the sgRNAs corresponding to each of these 10 genes, and then the cells were treated with 30 ng/mL mycolactone and assessed for viability. THP-1 cells transduced with an sgRNA against *EGFP* served as the negative control. Among the top 10 candidate genes, only cells with the *SEC61A1* knockout showed substantial prolongation of survival following mycolactone treatment, even after 144 h (Fig 3), which agrees with the exceptionally high RRA value (p = $7.46 \times 10^{-5}$) for *SEC61A1*, which was the highest among the 10 genes (Table 1). SEC61A1, the alpha 1 subunit of Sec61, is a component of the translocon, a transmembrane channel involved in the translocation of proteins across the endoplasmic reticulum (ER) membrane. Thus, SEC61A1 located on the ER membrane is a potential mediator of mycolactone-induced cell death, an essential pathologic characteristic of BU.

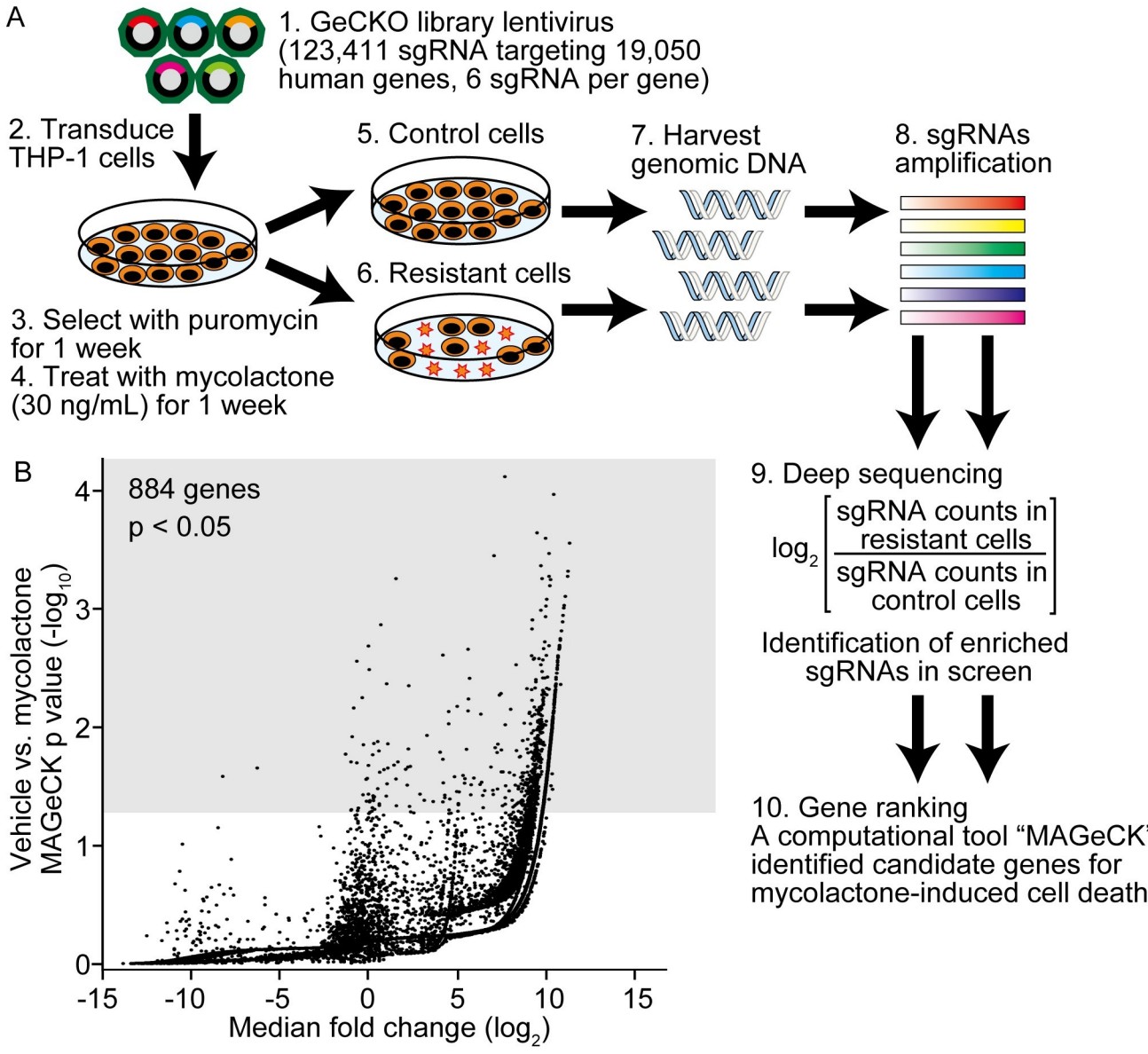

**Fig 2. GeCKO screening identified essential factors mediating mycolactone-induced cell death.** (A) Schematic representation of the GeCKO screening procedure (see Methods for details). (B) Volcano plot showing genes associated with mycolactone-induced cell death. The plot displays the $\log_2$ fold changes of the mean sgRNA counts (sgRNA count in resistant cells / sgRNA count in control cells) on the x-axis and $-\log10$ p-values from MAGeCK RRA comparisons on the y-axis. Genes with $p < 0.05$ are shown in the gray area of the plot.

## SEC61A1 mediates mycolactone-induced apoptosis

We next explored the possibility that SEC61A1 is involved in mycolactone-induced apoptosis, especially focusing on the molecular mechanisms of apoptosis in THP-1 cells. First, we confirmed SEC61A1 depletion by Western blotting, PCR analysis and DNA sequencing of the sgRNA binding site in *SEC61A1* (S2A–S2C Fig, respectively). Thereafter, we confirmed *SEC61A1* deletion by the CRISPR/Cas9 genome editing system. SEC61A1 is known to be important for protein translocation and calcium leakage, and SEC61A1 blockade affects cell growth and survival [34, 35]. After puromycin selection of knockout cells, we compared cell

**Table 1. The top 10 ranking genes identified by MAGeCK screening in mycolactone-treated cells.**

| Rank | Gene | Gene ID | Number of sgRNAs | Positive RRA value (p) |
|:---:|:---:|:---:|:---:|:---:|
| 1 | SEC61A1 | 29927 | 3 | $7.46 \times 10^{-5}$ |
| 2 | ZNF645 | 158506 | 2 | $1.05 \times 10^{-4}$ |
| 3 | IPO5 | 3843 | 3 | $2.23 \times 10^{-4}$ |
| 4 | CLEC12A | 160364 | 2 | $2.49 \times 10^{-4}$ |
| 5 | SULT1A3 | 6818 | 1 | $2.73 \times 10^{-4}$ |
| 6 | R3HDML | 140902 | 2 | $3.34 \times 10^{-4}$ |
| 7 | RNF141 | 50862 | 2 | $3.48 \times 10^{-4}$ |
| 8 | TNFRSF11B | 4982 | 3 | $4.25 \times 10^{-4}$ |
| 9 | ANKRD33 | 341405 | 1 | $4.72 \times 10^{-4}$ |
| 10 | SGSH | 6448 | 2 | $5.22 \times 10^{-4}$ |

Summary of the top 10 ranking candidate genes identified by MAGeCK screening. Each Gene ID was provided by the NCBI database. "Number of sgRNAs" refers to the number of significantly enriched sgRNAs corresponding to each target gene.

growth in control, *EGFP*- and *SEC61A1*-knockout cells (S2D Fig). *SEC61A1*-knockout cells grew more slowly than intact THP-1 cells and *EGFP*-knockout cells.

Next, we used FLICA, which is a fluorescent cell-permeable probe that binds selectively to activated caspase-3/7. Mycolactone increased the level of apoptosis compared with no treatment in *EGFP*-knockout cells, but this increase was prevented in *SEC61A1*-knockout cells (Fig 4A and 4B). However, there was no difference in the proportion of apoptotic cells between *EGFP*- and *SEC61A1*-knockout cells treated with ActD (Fig 4A and 4B). Western blot analysis revealed that mycolactone increased the cleaved caspase-3 level in control THP-1 cells, while caspase-3 cleavage was not detected in *SEC61A1*-knockout cells (Fig 4C). ActD is a

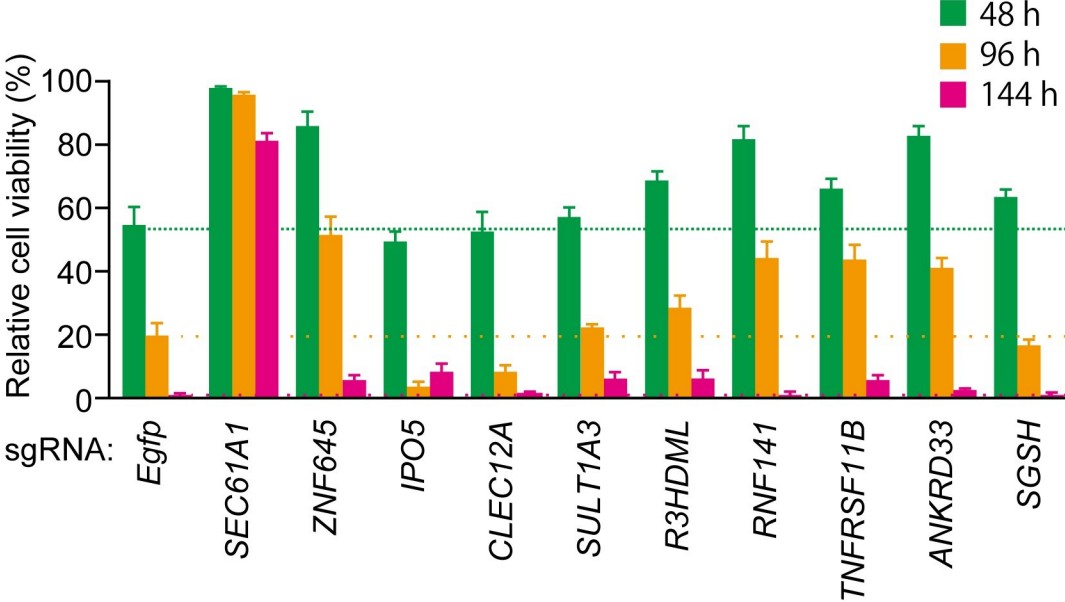

**Fig 3. Viability of sgRNA-transduced THP-1 cells treated with mycolactone.** The top 10 sgRNAs identified by MAGeCK screening were introduced into THP-1 cells. Then, the cells were treated with mycolactone at 30 ng/mL for 48, 96 and 144 h, after which cell viability was determined by the trypan blue exclusion assay (n = 5). Data are expressed as the cell viability relative to that of non-treated *EGFP*-knockout cells. *: p < 0.05; **: p < 0.01; ***: p < 0.005.

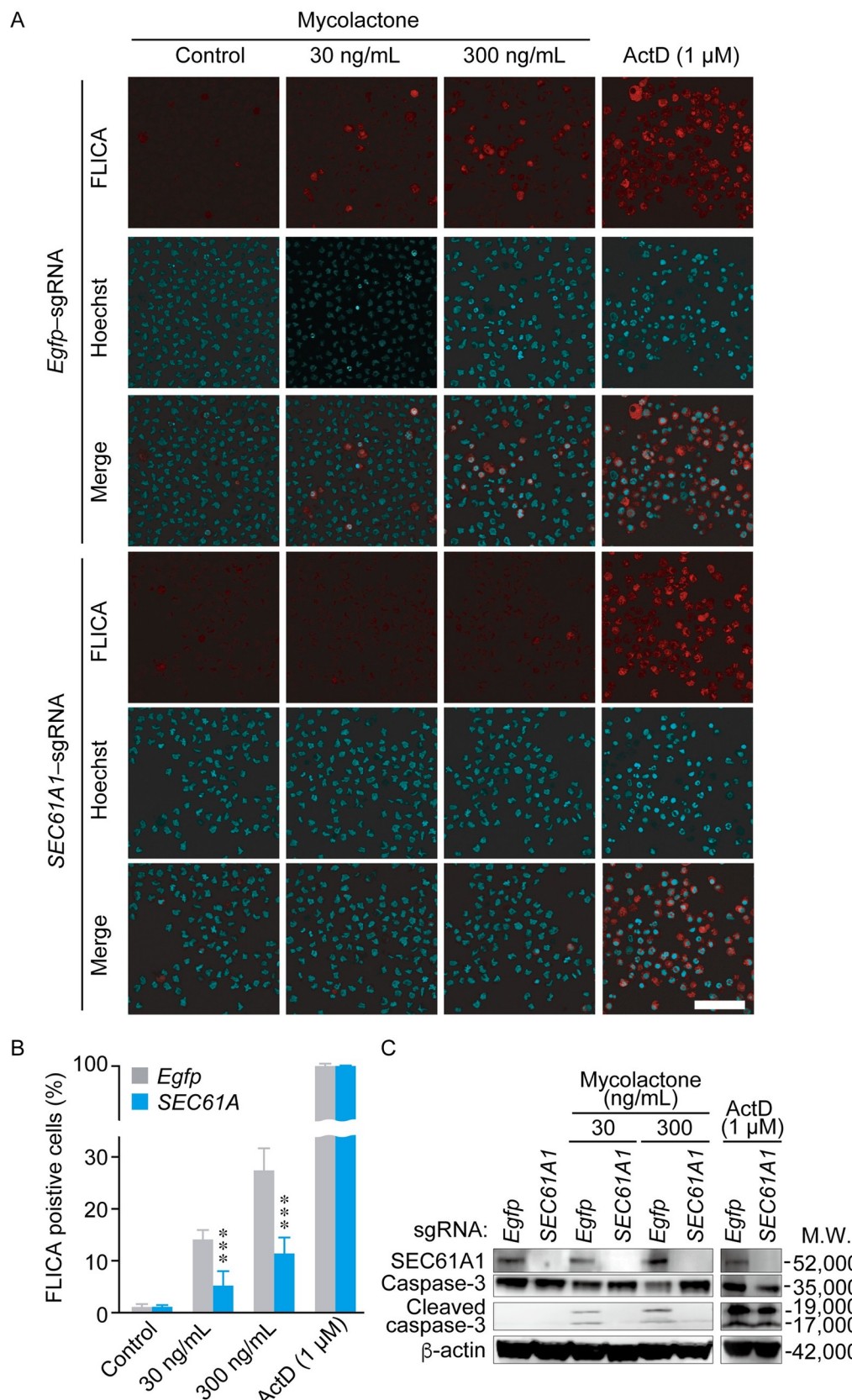

**Fig 4. Mycolactone-induced caspase-3-dependent apoptosis was suppressed by knockout of *SEC61A1*.** (A) Caspase activation assessed by FLICA assay. Cells were treated with 30 or 300 ng/mL mycolactone for 48 h. Caspase-3/7 activity was visualized by incubation with a fluorescent cell-permeable probe (FLICA) that binds selectively to activated caspase-3/7 in apoptotic cells. FLICA-positive cells (red) were observed using the FV10i confocal laser scanning microscope. Scale bar: 50 μm. (B) Quantitation of FLICA-positive cells (n = 6). ***: p < 0.005. (C) Western blot analysis of caspase-3 activation. Proteins from control (*EGFP*) and *SEC61A1*-knockout (*SEC61A1*) THP-1 cells treated with 30 or 300 ng/mL mycolactone for 48 h were subjected to Western blot analysis using antibodies against SEC61A1, caspase-3 and cleaved caspase-3. β-actin was used as a loading control. ActD was used as a positive control of caspase-3-dependent apoptosis.

transcription inhibitor which intercalates into DNA to inhibit new mRNA synthesis, arrest the cell cycle and induce caspase-3 dependent apoptosis [36], independent of ER stress. As a result, the knockout of *SEC61A1* had no effect on ActD-induced caspase-3 activation following mycolactone treatment (Fig 4C). These results suggest that *SEC61A1* is essential for mycolactone-mediated caspase-3 activation, but not for ActD-induced apoptosis in human monocytic THP-1 cells.

## Mycolactone-induced ER stress response and proapoptotic gene expression was abolished by *SEC61A1*-knockout THP-1 cells

SEC61A1 plays a key role in protein transport and calcium signaling in the ER membrane [37]. Dysfunction of SEC61A1 leads to accumulation of misfolded proteins, thereby activating ER-stress-related genes, which in turn activate proapoptotic genes [15, 38]. We therefore evaluated the mRNA expression of genes related to ER stress after mycolactone treatment in THP-1 cells using real-time PCR. In control (*EGFP*-knockout) cells, mycolactone significantly induced the mRNA expression of activating transcription factor 4 (*ATF4*) and DNA damage inducible transcript 3 (*DDIT3*), which are early response ER-stress-related genes; however, this increased expression was completely abolished in *SEC61A1*-knockout cells (Fig 5A). In addition, the mRNA levels of the proapoptotic genes phorbol-12-myristate-13-acetate-induced protein 1 (*PMAIP1*), B-cell lymphoma 2 binding component 3 (*BBC3*) and B-cell lymphoma 2 like 11 (*BCL2L11*) were significantly induced by mycolactone treatment in *EGFP*-knockout cells, but not in *SEC61A1*-knockout cells (Fig 5B). mRNA expression of *ATF4* and *DDIT3* was detected at 6 h after mycolactone treatment, while that of *BBC3* and *BCL2L11* was detected at much later time points (S3 Fig) [38].

The ER stress response is triggered through three pathways, *i.e.*, IRE1-*X-box binding protein 1 (XBP1)* splicing, PERK-eIF2α phosphorylation and ATF6 cleavage [38, 39]. When ER stress is activated, IRE1α splices the *XBP1* mRNA, producing an active transcription factor that stimulates the ER stress response [38, 39]. We therefore performed real-time PCR to evaluate the mRNA levels of these genes. *XBP1* splicing was induced in 3 h, peaked at 6 h and decreased in 24 h in *EGFP*-knockout cells (Fig 5C), suggesting that mycolactone activated the IRE1α-*XBP1* pathway. However, *SEC61A1*-knockout cells showed higher levels of *XBP1* splicing even before mycolactone treatment (Fig 5C).

Next, we analyzed the phosphorylation of proteins involved in the ER stress pathway by Western blotting. The eIF2α phosphorylation was induced by mycolactone in *EGFP*-knockout cells, but not in *SEC61A1*-knockout cells (Fig 5D). The PERK protein showed a weak signal, and its phosphorylation was not evident in THP-1 cells. Although full-length ATF6 was not detected in THP-1 cells, thapsigargin, an inducer of ER stress, induced ATF6 in *EGFP*-knockout cells but not in *SEC61A1*-knockout cells. Thapsigargin also induced IRE1α protein synthesis and eIF2α phosphorylation in all these cells. This evidence suggests that *SEC61A1* knockout affects ATF6 protein synthesis. These results also indicate that mycolactone induced *XBP1* splicing and eIF2α phosphorylation in THP-1 cells. Cells with the *SEC61A1* deletion

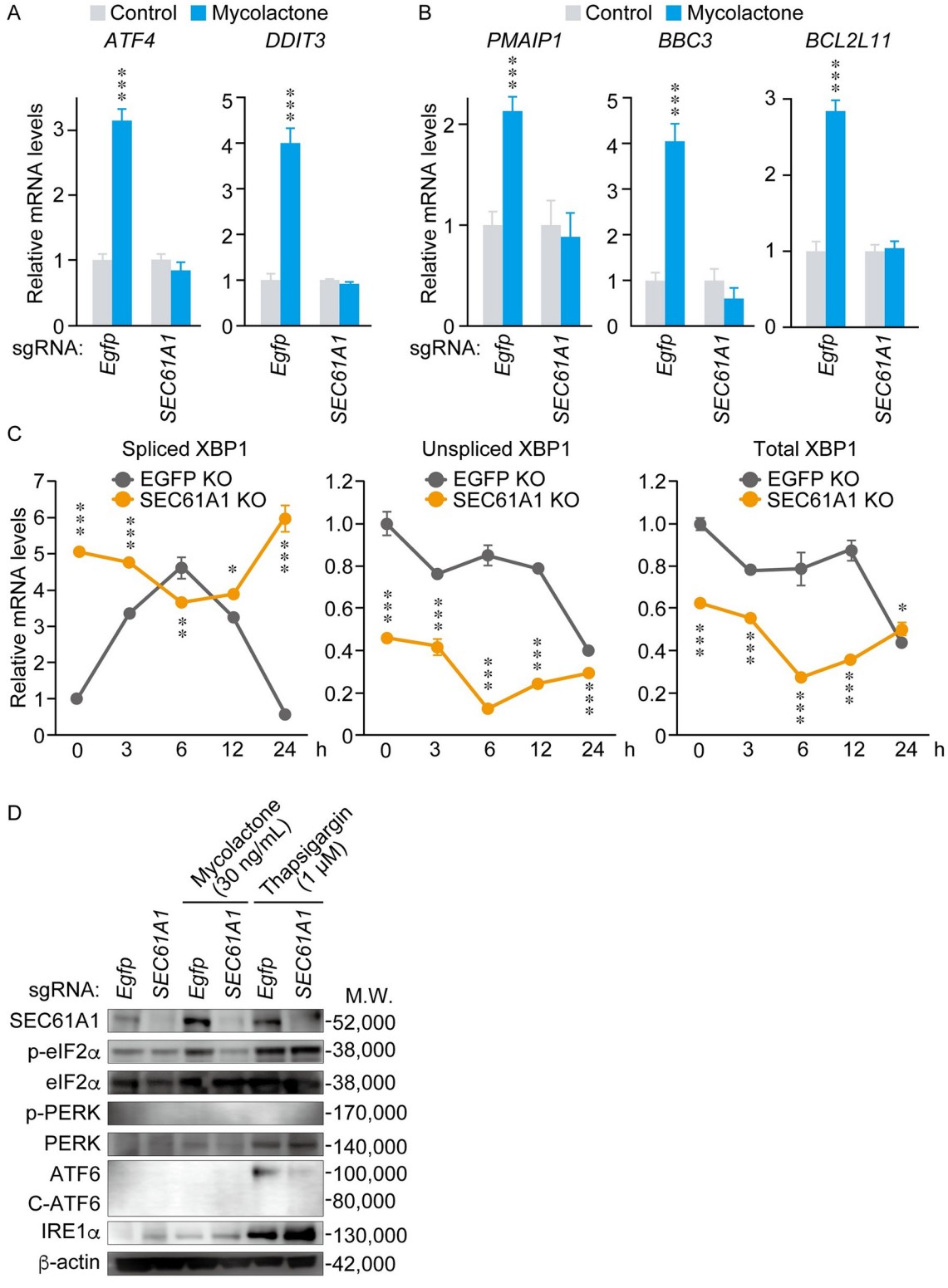

**Fig 5. Knockout of *SEC61A1* suppressed the mycolactone-induced ER stress response and expression of proapoptotic genes.**
Control *EGFP*-knockout and *SEC61A1*-knockout THP-1 cells were treated with 30 ng/mL mycolactone for 6 h (*ATF4*, *DDIT3* and *PMAIP1*) or 48 h (*BBC3* and *BCL2L11*). (A) The mRNA levels of ER-stress-related genes (*ATF4* and *DDIT3*) and (B) proapoptotic genes (*PMAIP1*, *BBC3* and *BCL2L11*) were analyzed by real-time RT-PCR. mRNA levels are expressed relative to those of ethanol-treated *EGFP*-knockout or *SEC61A1*-knockout THP-1 cells (n = 3). (C) The mRNA levels of unspliced, spliced and total *XBP1* using

control *EGFP*-knockout and *SEC61A1*-knockout THP-1 cells treated with 30 ng/mL mycolactone for 0, 3, 6, 12, and 24 h. mRNA levels are expressed relative to those of mycolactone-treated *EGFP*-knockout THP-1 cells (n = 3). *: p < 0.05; **: p < 0.01; ***: p < 0.005. (D) Control *EGFP*-knockout and *SEC61A1*-knockout THP-1 cells were treated with 30 ng/mL mycolactone for 24 h. ER stress proteins (p-eIF2α, eIF2α, p-PERK, PERK, full length ATF6, cleaved C-ATF6, and IRE1α) and SEC61A1 were evaluated by Western blotting. 1 μM thapsigargin for 24 h was included as a positive control for ER stress.

showed sustained *XBP1* splicing but were resistant to mycolactone treatment (Figs 5C and S2D). Taken together, these results suggest that *SEC61A1* is an essential factor mediating the ability of mycolactone to induce eIF2α phosphorylation.

## Discussion

Refractory skin ulceration caused by mycolactone, an exotoxin of *M. ulcerans*, is the main pathologic feature of BU and a potentially valuable therapeutic target. However, the mechanisms underlying the induction of cell death by mycolactone are not understood. Using GeCKO screening, we identified SEC61A1 as an essential factor mediating mycolactone induction of caspase-3-dependent apoptosis. Thus, our results corroborate previous studies using mutation screening and structural analysis, which identified SEC61A1 as one of the targets of mycolactone action [11, 13, 15, 40]. Since SEC61A1 is a component of the ER translocon, we additionally showed that mycolactone activates the expression of ER stress-related genes, which are known to trigger cytochrome c release from mitochondria to activate the caspase cascade and thus apoptosis [41]. We showed that SEC61A1 is specific for apoptosis induced by mycolactone, but not that induced by ActD, a well-known transcriptional inhibitor that triggers nucleolar stress and ultimately apoptosis [41].

SEC61A1 is a component of the Sec61 translocon complex responsible for the transportation of signal peptide precursors across the ER membrane to newly synthesized proteins [42]. The Sec61 complex also functions as an ER $Ca^{2+}$ channel that modulates calcium homeostasis, $Ca^{2+}$ is required for protein folding in the ER and its leakage will lead destabilization of proteins [43]. In a clinical study, *de novo* missense mutations in *SEC61A1* were reported to be the cause of common variable immunodeficiency and glomerulocystic kidney disease, a rare hereditary disorder characterized by the cystic dilation of Bowman's capsule due to protein instability and functional impairment with dysregulated calcium homeostasis [37]. In addition, other inhibitors that directly target SEC61A1, such as CT7, CT8, and ipomoeassin F, have similar effects to mycolactone [8–12]. Mycolactone physically binds to SEC61A1, maintains the Sec61 translocon conformation, and prevents the access of signaling peptides to the binding site of SEC61A1 [13]. Mycolactone alters the Sec61 translocon conformation to open the cytosolic side of the lateral gate and enhance calcium leakage [44]. Mycolactone also prevents the translocation of proteins that pass through the endoplasmic reticulum for secretion or placement in cell membranes [7, 8, 45]; these proteins accumulate and activate ER stress responses [7, 46]. In the present study, we demonstrated that the knockout of *SEC61A1* blocked the mycolactone-induced expression of ER stress-related genes, eIF2α phosphorylation, and apoptosis in human premonocytic THP-1 cells. Therefore, mycolactone directly binds to SEC61A1 thereby inhibiting the protein translocation into the ER and enhancing calcium leakage, which results in the accumulation of misfolded mislocated proteins [8, 46, 47]. This process causes ER stress and subsequently activates caspase-3 to induce apoptosis in affected cells. However, so far the relationship between ER stress and oxidative stress was not confirmed. Mycolactone also induces reactive oxygen species, which affect cytotoxicity [48, 49].

SEC61A1 has an important role in protein translocation and calcium leakage [42, 46]. It was reported that *SEC61A1* deletion or silencing resulted in growth defects or cell death in

association with sustained *XBP1* splicing [34, 35, 50]. In our study, deletion of *SEC61A1* showed a low growth phenotype and sustained *XBP1* splicing in THP-1 cells (Figs 5C and S2D). However, *SEC61A1* deletion did not induce apoptosis or the expression of proapoptotic genes such as *PMAIP1*, *BBC3* and *BCL2L11* (Fig 5B). We could not elucidate the mechanism contributing to the different growth and cytotoxic phenotype of *SEC61A1* deletion between THP-1 cells and other cells. In our genome-wide screening, *SEC61A1*-knockout cells had low growth rates but could survive in the presence of mycolactone; *SEC61A1* was found as a top score gene (Table 1).

In the present genome-wide screening, several other genes were identified as candidates involved in mycolactone-induced cell death, although their RRA values were much lower than that of *SEC61A1* (Table 1). These genes include the C-type lectin domain family 12 member A (*CLEC12A*), which encodes an early adaptor molecule involved in autophagy to eliminate stress-related proteins [51], and the ring finger protein 141 (*RNF141*), a gene that is upregulated in the jejunum of *XBP1*-silenced mice [52]. However, the knockout of each of these genes did not alter mycolactone-induced apoptosis in THP-1 cells. Several other genes have been reported as candidates mediating mycolactone-initiated cell death, such as angiotensin II receptor type 2, WASP-like actin nucleation promoting factor, FKBP prolyl isomerase 1A, AKT serine/threonine kinase 2, mechanistic target of rapamycin kinase and SET domain containing 1B [14, 53–55]. However, none of these genes were detected as candidates in our GeCKO screening. Although these differences may be due to the different cell types used in each study [31], further analysis may be required to elucidate the whole picture of mycolactone-induced ER stress and apoptosis.

Mycolactone induced the ER stress response, which triggers cell apoptosis [15, 46]. ER stress responses are activated by three major pathways including IRE1-*XBP1* splicing, phosphorylation of eIF2α via multiple kinases (PERK, general control non-repressed 2: GCN2, protein kinase RNA-activated: PKR), and ATF6 cleavage. Each pathway activates a pro-survival response, responsible for restoring normal ER function by reducing unfolded proteins [56]. Finally, failure to remove stress results in the expression of proapoptotic signaling genes such as *PMAIP1*, *BBC3* and *BCL2L11*.

In this paper, we showed that mycolactone induced IRE1-*XBP1* splicing and eIF2α phosphorylation, but not ATF6 cleavage. It was reported that mycolactone induced *XBP1* splicing in MutuDCs [46] and eIF2α phosphorylation in HeLa cells and MEF cells [15]. We also demonstrated that mycolactone induced ER stress responses in THP-1 cells. The *SEC61A1* deletion cells have diminished eIF2α phosphorylation and expression of proapoptotic genes. Although phosphorylation of PERK, one of the eIF2α kinases, was not detected in THP-1 cells, other kinases such as GCN or PKR may phosphorylate eIF2α with mycolactone treatment [15]. In the present study, the ER stress inducer thapsigargin induced eIF2α phosphorylation in *EGFP*- and *SEC61A1*-knockout THP-1 cells. ATF6 was upregulated by thapsigargin in control, but not in *SEC61A1*-knockout cells. Therefore, it is possible to infer that thapsigargin induces eIF2α phosphorylation independent of SEC61A1, whereas ATF6 expression was dependent on SEC61A1. It was reported that the SEC61A1 inhibitor ipomoeassin F also suppressed ATF6 protein expression in HepG2 cells [9]. Morel *et al*. demonstrated that mycolactone induced an atypical ER stress response, in which the ATF4/DDIT3 branch of the ER stress response was robustly activated, but *Bip*, an upstream regulator of the ER stress response, was downregulated [39, 46]. It was different from the conventional ER stress response induced by thapsigargin, tunicamycin, and MG132. Our study also showed mycolactone-induced *ATF4/DDIT3* expression, which was diminished by *SEC61A1* knockout. *ATF6* gene activation was induced only by thapsigargin and not by mycolactone. We have confirmed that the SEC61A1 knockout suppressed mycolactone-induced ER stress and cytotoxicity. However, the viability assay and

FLICA assay demonstrated that mycolactone-induced apoptosis was not completely prevented by the SEC61A1 knockout. This evidence suggests that other mycolactone-induced stress pathways may also be involved in inducing apoptosis in THP-1 cells. Further study is required to demonstrate the precise molecular mechanisms of SEC61A1 in ER stress apoptosis.

Several approaches have been proposed to inhibit mycolactone activity, *e.g.*, the use of IgG antibody against mycolactone [57, 58], the inhibition of the mycolactone synthesis pathway with substrates (malonyl-CoA, methyl malonyl-CoA) [59] and the attenuation of cytotoxicity with antioxidants [48, 49]. At present there is no effective drug to reduce the toxicity of mycolactone, and further drug screening and analysis of the biological activity of mycolactone will be required to develop novel therapeutic modalities to inhibit mycolactone-induced ulceration.

In summary, the present genome-wide study of GeCKO-screened THP-1 cells identified *SEC61A1* as an essential gene responsible for mycolactone-induced ER stress and caspase-3-dependent apoptosis. Since mycolactone is the sole pathologic factor of *M. ulcerans*, inhibition of mycolactone activity or its downstream cascades could be a novel therapeutic modality to eliminate the harmful effects of mycolactone, in addition to the 8-week antibiotic regimen of rifampicin and clarithromycin.

## Supporting information

**S1 Fig. Ranking of the candidate genes identified by GeCKO screening.** Genes were ranked according to the p-value of MAGeCK screening. Lower RRA p-values indicate a stronger positive selection of the corresponding gene. The top 10 ranked genes are labeled with red dots. Genes with a p <0.05 are shown in the gray area. Source data are provided in S1 Data. (DOCX)

**S2 Fig. *SEC61A1*-deleted THP-1 cells were generated using the CRISPR/Cas9 genome editing system.** (A) Cell lysates were prepared from control *EGFP*-knockout and *SEC61A1*-knockout THP-1 cells and analyzed by immunoblotting with Abs against SEC61A1. β-Actin was used as a loading control. (B) PCR amplification of genomic DNA from control *EGFP*-knockout and *SEC61A1*-knockout THP-1 cells, using a FWD primer targeting the *SEC61A1* sgRNA binding site and a BWD primer targeting 100 bp downstream. The higher PCR annealing temperature was set to 65˚C for 15 cycles. (C) Sequences of the *SEC61A1* sgRNA targeting site from PCR products, which were amplified from genomic DNA of control *EGFP*-knockout and *SEC61A1*-knockout THP-1 cells. The sequences are compared to the reference human genome sequence. N means that the nucleotide was not determined by sequencing due to multiple mutations inserted near the PAM sequence (protospacer adjacent motif, which is a CCG DNA sequence). (D) Cell proliferation assay. The cell proliferation assay indicated that control, *EGFP*-knockout and *SEC61A1*-knockout THP-1 cell numbers were distinctly increased in a time-dependent manner during the 5-day culture. The cell number and viability were measured by the trypan blue exclusion assay using an automatic cell counter (n = 6). ***: p < 0.005 compared with the cell number of control THP-1 cells. [†]***: p < 0.005 compared with the cell number of *EGFP*-knockout THP-1 cells. (DOCX)

**S3 Fig. Mycolactone induced the expression of ER-stress-related genes and proapoptotic genes.** The THP-1 cells were treated with 30 or 300 ng/mL mycolactone for the indicated periods. mRNA levels were analyzed by real-time RT-PCR and normalized to those of *GAPDH*. The expression levels of ER-stress-related genes (*ATF4* and *DDIT3*) and pro-apoptotic genes (*PMAIP1*, *BBC3* and *BCL2L11*) are relative to the pre-treatment levels (0 h) in the bar graph

(n = 3). $^*$: p < 0.05; $^{**}$: p < 0.01; $^{***}$: p < 0.005.
(DOCX)

**S4 Fig. Original uncropped images of Western blots from Figs 1, 4 and 5.** Molecular weight markers were included when available.
(DOCX)

**S1 Table. sgRNA sequences for each target gene.**
(DOCX)

**S2 Table. Primers used for real-time PCR.**
(DOCX)

**S1 Data. Source data file.** Ranking of genes identified by MAGeCK screening after treatment with mycolactone from the GeCKO v2 library.
(XLSX)

## Acknowledgments

We are grateful to Dr. Yoshito Kishi of Harvard University for the kind donation of synthetic mycolactone A/B. Anti-PERK and anti-p-PERK antibody were kindly provided by Dr Gen-ichi Atsumi of Teikyo University.

## Author Contributions

**Conceptualization:** Akira Kawashima, Junichiro En, Masamichi Goto, Koichi Suzuki.

**Data curation:** Akira Kawashima, Koichi Suzuki.

**Formal analysis:** Akira Kawashima.

**Funding acquisition:** Akira Kawashima, Keiji Maruyama, Shigekazu Watanabe, Masamichi Goto, Koichi Suzuki.

**Investigation:** Akira Kawashima.

**Methodology:** Akira Kawashima, Mitsuo Kiriya, Junichiro En, Kazunari Tanigawa, Yoko Fujiwara, Yuqian Luo, Masamichi Goto, Koichi Suzuki.

**Project administration:** Akira Kawashima.

**Resources:** Akira Kawashima, Junichiro En, Masamichi Goto, Koichi Suzuki.

**Software:** Akira Kawashima.

**Supervision:** Koichi Suzuki.

**Validation:** Akira Kawashima.

**Visualization:** Akira Kawashima, Koichi Suzuki.

**Writing – original draft:** Akira Kawashima.

**Writing – review & editing:** Mitsuo Kiriya, Junichiro En, Kazunari Tanigawa, Yasuhiro Nakamura, Yoko Fujiwara, Yuqian Luo, Koichi Suzuki.

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
