## [Decision Letter · Decision Letter 0]

6 Jan 2022

Dear Dr. Suzuki,

Thank you very much for submitting your manuscript "Genome-wide screening identified SEC61A1 as an essential factor for mycolactone-dependent apoptosis in human premonocytic THP-1 cells" for consideration at PLOS Neglected Tropical Diseases. As with all papers reviewed by the journal, your manuscript was reviewed by members of the editorial board and by several independent reviewers. In light of the reviews (below this email), we would like to invite the resubmission of a significantly-revised version that takes into account the reviewers' comments. 

Your research was generally well-received by the reviewers who agreed that, despite a now well-established literature on Sec61a as the target of mycolactone, this manuscript does provide additional and important novel insights. However, all reviewers had similar concerns about some aspects of the experimental data, and the conclusions drawn. The reviews are broadly in agreement, but the additional guidance in the preparation of the revision is provided:

All reviewers were concerned about the viability of THP-1 cells in which SEC61A1 had been knocked out by CRISPR. It is essential that experimental data directly comparing viability of SEC61A1 knockout cells to control cells (preferably non-targeting sgRNA) in the absence and presence of mycolactone at different timepoints is presented. You may need to carefully consider the normalisation strategy to ensure this data accurately reflects the findings. It should also be confirmed whether the knockout is complete or partial (either by PCR of targeted cell genomes, or by testing the ability of these cells to make Sec61-dependent proteins such as cytokines).

The reviewers felt that the your findings of transcriptional responses in several genes were not sufficient to specifically implicate ER stress in the response to mycolactone. Here, you have the option of either providing additional experimental data to support the induction of ER stress (activation of all three arms: XBP-1, ATF6 and eIF2a), or changes to the abstract/keywords/manuscript text to reflect the existing literature taking all reviewers’ comments into account. Please also clarify the point identified by reviewer 1, regarding XBP-1 (I believe this refers to lines 428-431)

In terms of other changes to the text, two critical areas are highlighted by the reviewers.

First, the final conclusion that Sec61 is a therapeutic target in Buruli ulcer is not justified, since all Sec61 inhibitors have a similar toxicity profile to mycolactone. Please change these aspects in line with the reviewers’ comments. 

Second, it is important that the prior literature on this topic is cited; there are a significant number of specific areas and papers suggested by each reviewer, and a Pubmed search of Sec61 and mycolactone may reveal more. The discussion should place your work in the context of these previous findings. I would be particularly interested to hear your thoughts on how the toxicity profile of mycolactone and/or Sec61 depletion may have a different profile in the non-adherent THP-1 cells to adherent cells investigated by others.

We cannot make any decision about publication until we have seen the revised manuscript and your response to the reviewers' comments. Your revised manuscript is also likely to be sent to reviewers for further evaluation.

Sincerely,

Rachel E Simmonds, Ph.D.

Guest Editor

Gerd Pluschke

Deputy Editor

Your research was generally well-received by the reviewers who agreed that, despite a now well-established literature on Sec61a as the target of mycolactone, this manuscript does provide additional and important novel insights. However, all reviewers had similar concerns about some aspects of the experimental data, and the conclusions drawn. The reviews are broadly in agreement, but the additional guidance in the preparation of the revision is provided:

All reviewers were concerned about the viability of THP-1 cells in which SEC61A1 had been knocked out by CRISPR. It is essential that experimental data directly comparing viability of SEC61A1 knockout cells to control cells (preferably non-targeting sgRNA) in the absence and presence of mycolactone at different timepoints is presented. You may need to carefully consider the normalisation strategy to ensure this data accurately reflects the findings. It should also be confirmed whether the knockout is complete or partial (either by PCR of targeted cell genomes, or by testing the ability of these cells to make Sec61-dependent proteins such as cytokines).

The reviewers felt that the your findings of transcriptional responses in several genes were not sufficient to specifically implicate ER stress in the response to mycolactone. Here, you have the option of either providing additional experimental data to support the induction of ER stress (activation of all three arms: XBP-1, ATF6 and eIF2a), or changes to the abstract/keywords/manuscript text to reflect the existing literature taking all reviewers’ comments into account. Please also clarify the point identified by reviewer 1, regarding XBP-1 (I believe this refers to lines 428-431)

In terms of other changes to the text, two critical areas are highlighted by the reviewers.

First, the final conclusion that Sec61 is a therapeutic target in Buruli ulcer is not justified, since all Sec61 inhibitors have a similar toxicity profile to mycolactone. Please change these aspects in line with the reviewers’ comments. 

Second, it is important that the prior literature on this topic is cited; there are a significant number of specific areas and papers suggested by each reviewer, and a Pubmed search of Sec61 and mycolactone may reveal more. The discussion should place your work in the context of these previous findings. I would be particularly interested to hear your thoughts on how the toxicity profile of mycolactone and/or Sec61 depletion may have a different profile in the non-adherent THP-1 cells to adherent cells investigated by others.

Reviewer's Responses to Questions

**Key Review Criteria Required for Acceptance?**

**Methods**

-Are the objectives of the study clearly articulated with a clear testable hypothesis stated?

-Is the study design appropriate to address the stated objectives?

-Is the population clearly described and appropriate for the hypothesis being tested?

-Is the sample size sufficient to ensure adequate power to address the hypothesis being tested?

-Were correct statistical analysis used to support conclusions?

-Are there concerns about ethical or regulatory requirements being met?

Reviewer #1: No major new analyses/experiments are required to make the study acceptable for publication.

I have, however, suggested two additional experiments that would enhance the novelty of the study and further our understanding of the mechanism of mycolactone-induced cell death in THP-1 cells (see final paragraph of the Summary and General Comments section). Both suggestions are potentially beyond the scope of the current study so, provided that the authors address the points that I have listed in the 'Editorial and Data Presentation Modifications' section, I recommend that the study is acceptable for publication with 'minor revisions', even if the suggested experiments are not performed. This is because the current methods and analyses answer the stated objectives, have been performed with scientific rigour and used correctly to support the current conclusions.

Reviewer #2: The methods are fully described and appropriate to the objectives of the study. The statistics are sufficient to support the conclusions. There are no ethical or regulatory concerns.

Reviewer #3: The work was carried out with the appropriate methodology.

**Results**

-Does the analysis presented match the analysis plan?

-Are the results clearly and completely presented?

-Are the figures (Tables, Images) of sufficient quality for clarity?

Reviewer #1: Presented analysis matches the analysis plan.

Results are clearly and completely presented and figures are clear. Minor errors: a spelling mistake in Figure 2 and the implication that XBP1 was identified as candidate in the text but not in any of the Figures, Table 1 or source data (more detail given in the Editorial and Data Presentation Modifications section).

Reviewer #2: The data are very well presented and all figures and legends are of high quality.

Reviewer #3: I ma happy with presentation of the results.

**Conclusions**

-Are the conclusions supported by the data presented?

-Are the limitations of analysis clearly described?

-Do the authors discuss how these data can be helpful to advance our understanding of the topic under study?

-Is public health relevance addressed?

Reviewer #1: Conclusions are supported by the presented data.

Limitations are described and the authors discuss how these data can advance our understanding of mycolactone-induced cell death but do not place this in the context of current literature where these mechanisms have been elucidated in significant detail using other cell lines (Ogbechi et al. 2018, Morel et al. 2018).

Public health relevance is briefly discussed but lacks clarity.

Reviewer #2: I have several reservations about the conclusions drawn from this data and feel more work is needed to support the authors' interpretation of the results, mostly related to conclusion that the cell death caused by mycolactone is entirely due to the induction of ER stress.

Reviewer #3: The conclusions are well supported by the data. However, limitations of the study are not currently sufficiently discussed especially with regard to the known role of Sec61 in mediating mycolactone cytotoxicity. This needs to be corrected in a revised manuscript.

**Editorial and Data Presentation Modifications?**

Reviewer #1: Line 118 vs. 247: Z-DEVD-FMK is described as ‘a specific and irreversible caspase-3 inhibitor’ (line 118) and a ‘pan-caspase inhibitor’ (line 247). Z-DEVD-FMK shows potent inhibition of other caspases so is not a ‘specific’ inhibitor of caspase-3 (please amend in line 118). Since Z-DEVD-FMK also inhibits caspase-8 (which acts upstream of caspase-9 and caspase-3) in cells, perhaps it would be more appropriate to say that ‘mycolactone induced caspase-dependent apoptosis in THP-1 cells’ (line 254) rather than being specific for caspase-3?

Line 301: spelling error in Fig. 2A, step 4; change ‘mycoractone’ to ‘mycolactone’. 

Line 330: Was there a ‘no treatment’ control for any of the sgRNA treated cells? Or have the values shown here been expressed relative to the ‘no treatment’ for each sgRNA? For example, I’d expect the viability of SEC61A1-KO cells to start declining after 96-144 h even when not treated with mycolactone? 

Line 336: Reference(s) for FLICA probe?

Line 340: Could benefit from stating that Actinomycin D (ActD) is a transcription inhibitor ie. to make it clearer that there is no difference in apoptosis induction between the EGFP- and SEC61A1-KO cells.

Line 401: Suggest text is amended to: ‘which are known to trigger cytochrome c release from…’ It currently sounds like this has been experimentally determined in this manuscript when it hasn’t.

Line 426-431: The authors include XBP1 as one of the genes identified as a candidate of mycolactone-induced cell death with a lower RRA value than SEC61A1 and state that knockout of these genes did not alter mycolactone-induced apoptosis in THP-1 cells. However, there is no mention of XBP1 in any of the Figures, Table 1 or source data.

Line 441: Note spelling error; change ‘dependant’ to ‘dependent’. 

Line 441-447: This section lacks clarity. What is meant by ‘inhibition of mycolactone’? Do you mean inhibit its biogenesis by M. ulcerans or something else? Could you expand on how Sec61alpha could be inhibited with reduced toxicity than that induced by mycolactone-mediated inhibition of Sec61? Do you think that a more substrate-selective Sec61 inhibitor (eg. the cotransin derivative CT8; see Pauwels et al. 2021) could be used to compete with mycolactone-Sec61 binding during an M. ulcerans infection?

Line 541: The authors acknowledge ‘synthetic and biotinylated mycolactone A/B’ but biotinylated mycolactone is not mentioned elsewhere in the manuscript?

Reviewer #2: There are some errors in interpretation of current knowledge that need to be corrected.

1. In line 407, the authors state that calcium leakage from the ER is “required for stabilization of protein folding” and in line 421 suggest that mycolactone inhibits calcium leakage. Calcium leakage is an inevitable consequence of the opening of the Sec61 channel. Countering this to maintain ER calcium levels is the process required for protein folding and cells go to some lengths to minimise the leak via BiP in the ER lumen and calmodulin in the cytosol. Recent evidence suggests that mycolactone does not inhibit the leak but actually increases it (see Bhadra et al, 2021, PMID: 34726690). This needs to be discussed.

2. In line 420 the authors speculate that “mycolactone directly binds to SEC61A1, thereby inhibiting modification of signalling peptides … which results in accumulation of unfolded proteins in the ER” and in line 446 they suggest that the effects of mycolactone may be blocked by inhibiting SEC61A1. It is now well established that mycolactone is in fact an inhibitor of SEC61A1 (see refs 29 and 30 for example). As mycolactone prevents proteins passing through the translocon, the authors need to address how it might cause accumulation of unfolded proteins within the ER. Finally, most of the other known inhibitors of SEC61A1 activity induce a remarkably similar phenotype to mycolactone (eg Ipomoeassin F, see Zong et al, 2019, PMID: 31059257 and Roboti et al, 2021, PMID: 34079010) and therefore would be expected to mimic the effects of mycolactone rather than protect against them. These parts of the discussion need to be rewritten.

Reviewer #3: I would recommend considering this manuscript if my concerns can be adequately addressec.

**Summary and General Comments**

Reviewer #1: The manuscript by Kawashima et al. focusses on understanding the genes necessary for mycolactone-induced cell death in human premonocytic THP-1 cells. Mycolactone is an exotoxin produced by Mycobacterium ulcerans, the causative agent of the necrotising skin disease known as Buruli ulcer, and it is well established that the pathology of this disease is directly linked to mycolactone binding to the central subunit of the Sec61 complex, Sec61alpha (Demangel and High 2018). The Sec61 complex is the predominant protein conducting channel via which secretory and transmembrane proteins traverse, or are inserted into, the membrane of the endoplasmic reticulum (ER) (O’Keefe et al. 2021). By binding to the cytosolic side of Sec61alpha and stabilising a partially open conformation of Sec61 (Gérard et al. 2020), mycolactone precludes the entry of most newly synthesised polypeptides into the Sec61 channel and, hence, inhibits their access to the secretory pathway. This global blockade in ER protein translocation reduces the ability of cells to synthesise many secretory and transmembrane proteins (type I, type II but not type III or tail-anchored proteins; McKenna et al. 2017, Morel et al. 2018, Zong et al. 2019), triggering stress responses that ultimately induce apoptosis and cell death (Ogbechi et al. 2018, Morel et al. 2018).

Here, the authors first use trypan blue and Hoescht 33342 staining, together with the pan-caspase inhibitor Z-DEVD-FMK, to show that mycolactone-induced cell death proceeds via a caspase-3 dependent apoptotic pathway in THP-1 cells. They then use genome-wide CRISPR/Cas9-mediated screening to identify 884 candidate genes involved in mycolactone-induced cell death. After sgRNA-mediated knockout of each of the top 10 scored candidate genes, the authors find that only SEC61A1-KO THP-1 cells showed prolonged survival when treated with mycolactone. Next, by using a fluorescent probe that binds to activated caspase-3, the small molecule apoptosis inducer Actinomycin D and western blot analysis, the authors show that Sec61alpha is required for mycolactone-induced caspase-3 activation. Finally, the authors use real-time PCR to show that mycolactone-treated THP-1 cells induce mRNA expression of the ER-stress related genes ATF4 and DDIT3 (CHOP) after 6 h and, after prolonged exposure, increased mRNA levels of the pro-apoptotic genes PMAIP1 (Noxa), BBC3 (Puma) and BCL2L11 (Bim); effects that are ablated in SEC61A1-KO cells. Taken together, the authors conclude that Sec61alpha is essential for the ability of mycolactone to induce ER stress and caspase-3 dependent apoptosis. From this, they propose that either inhibiting Sec61alpha or downstream ER stress and caspase-3 dependent apoptotic factors presents a novel approach to eliminate the harmful effects of mycolactone during M. ulcerans infection.

Overall, this is a well-executed and technically sound study. However, despite the use of sophisticated biochemical methods and genome-wide screening, most of what is shown here confirms previous work which has already established that mycolactone’s toxicity strictly depends on its binding to Sec61alpha (Baron et al. 2016) and that ATF4 and CHOP mRNA expression is increased in response to mycolactone in RAW264.7 (Ogbechi et al. 2018), HeLa (Ogbechi et al. 2018) and dendritic (Morel et al. 2018) cells. 

The present study modestly extends this work by demonstrating that, in THP-1 cells, mycolactone-treatment increases the mRNA levels of ATF4, CHOP, PMAIP1 (Noxa), BBC3 (Puma) and BCL2L11 (Bim) and induces caspase-3 dependent apoptosis. Whether this caspase-dependent apoptosis is the result of; i) a mycolactone-induced integrated stress response in the absence of the UPR (RAW264.7 and HeLa cells; Ogbechi et al. 2018); ii) an atypical UPR as observed in mycolactone-treated dendritic cells (Morel et al. 2018) and ipomoeassin-F-treated HepG2 cells (Roboti et al. 2021) or iii) a combination of stress responses in THP-1 cells, would certainly enhance our understanding of mycolactone-induced cell death in different cell types. The authors could potentially address this question by monitoring XBP1 mRNA splicing in mycolactone-treated THP-1 cells using real-time or reverse-transcription PCR (cf. Ogbechi et al. 2018, Morel et al. 2018, Roboti et al. 2021) or assessing the level of mRNA induction of ATF4 and/or CHOP in mycolactone-treated ATF4-KO THP-1 cells (cf. Ogbechi et al. 2018). Although these experiments are potentially beyond the scope of the current study, the authors can significantly improve their manuscript by placing their results within the context of other recent studies that have advanced our understanding of the mechanisms of mycolactone-induced apoptosis (eg. Morel et al. 2018, Ogbechi et al. 2018).

In summary, Kawashima et al. set out to identify the genes necessary for mycolactone-induced cell death in human premonocytic THP-1 cells. They successfully achieved their objective and, out of 884 candidate genes identified by genome-wide screening, they identified Sec61A1 as the only essential gene that is required for the mycolactone-mediated induction of caspase-dependent apoptosis in THP-1 cells.

Reviewer #2: This work describes an investigation into the mechanism of cell death induced by the Buruli Ulcer virulence factor mycolactone using genome scale CRISPR screen in THP-1 cells. The authors found and validated SEC61A1 as a candidate gene. Knockout of SEC61A1 protected cells against mycolactone-induced cytotoxicity, prevented caspase 3 activation and reduced expression of stress related and pro-apoptotic genes. This well written and interesting paper contributes to our understanding of mycolactone’s activity. However, as well as needing further experiments to back up the conclusions, there are some omissions in the document the weaken the overall impact. There is a brief discussion of the results in the context of previously published works but there are many highly relevant publications that have been largely ignored. For example, the widely reported role of oxidative stress in mycolactone cytotoxicity is not mentioned and multiple papers showing inhibition of Sec61 activity by mycolactone are uncited. When comparing their results to other screens, the authors merely suggest that the differences in their findings to previous screens may be due to cell type, but this seems unlikely given the universal expression of SEC61A1 and the essential nature of its activity. The discussion should be expanded to include a more thorough analysis of the literature and the implications of the findings. 

Experimentally there are several areas that need more work to justify the authors conclusions.

1. There are a number of CRISPR-based studies that have identified SEC61A1 as an essential gene and report a severe growth defect in knockout cells (eg Adamson et al, 2016, PMID: 27984733). Some data on the growth of the knockout THP-1 cells would be useful here as changes in growth may affect cellular responses. Confirmation of SEC61A1 knockout by PCR would also strengthen the authors findings.

2. The stress response genes analysed by the authors are not specific for ER-stress and could be induced by other pathways. It cannot be concluded that mycolactone induces ER-stress based on this alone. This is important because Ogbechi et al (Ref 24) showed that mycolactone can activate an integrated stress response via multiple pathways. In addition, Adamson et al (see above) reported that SEC61A1 knockout actually activates the IRE1/XPB1 arm of the ER-stress response. To make a strong case for ER-stress as the major cause of cell death in mycolactone treated cells, the authors need to show all three arms of the stress response are activated by demonstrating EIF2α phosphorylation, XBP1 splicing and ATF6 cleavage. Use of a control inducer of ER-stress that does not depend directly on Sec61 such Tunicamycin or Thapsigargin would help confirm that the protective effect of SEC61A1 knockout is specific for mycolactone. 

3. Related to the above point, the activators of the unfolded protein response, PERK, IRE1 and ATF6 are all Sec61 dependent proteins. These proteins could be downregulated in the knockout THP-1 cells prior to addition of mycolactone, which would explain the reduced response. The levels of each should be confirmed by immunoblotting.

Reviewer #3: Review of manuscript of Kawashima et al. entitled ” Genome-wide screening identified SEC61A1 as an essential factor for mycolactone-dependent apoptosis in human premonocytic THP-1 cells”

This manuscript describes a CRISPR screen to identify factors involved in mycolactone-induced apoptosis in premonocytic THP-1 human cells. The screen identified SEC61A1 as the main hit gene, whose downmodulation was further validated to confer protection from mycolactone-induced cytotoxicity. It is already well established that mycolactone directly binds to the trimeric Sec61 translocon and blocks its protein translocation activity. Since Sec61 is essential for viability of all eukaryotic cells, its inhibition by mycolactone leads to broad cytotoxicity across many mammalian cell lines. The screen described in this manuscript is well carried out and the SEC61A1 hit is validated and new insights into the apoptosis mechanism resulting from Sec61 inhibition are presented. The already well-established role of Sec61 as the direct cellular target of mycolactone somewhat detracts from the general impact of this work. Still, this work is worth publishing given that the below comments are addressed by the authors.

Major points:

1) It is already quite abundantly demonstrated by previous studies that the direct cellular target of mycolactone A/B responsible for its cytotoxic effects is the Alpha subunit of the Sec61 complex. I believe it is not correct to completely omit this information from the abstract and this should be corrected in the revised manuscript.

2) Introduction p. 5. Here, the authors state that the essential host factors that mediate the action of mycolactone remain largely unknown. In fact, both the Demangel and Simmonds laboratories have demonstrated that in mammalian cells, the direct target of mycolactone responsible for the compound cytotoxicity is the Alpha subunit of the Sec61 complex. The first papers in this are (PMID: 24699819, 27821549). It is required that this previous determination of the cellular target of mycolactone is at least mentioned and the relevant literature is cited.

3) If I understand it correctly, the CRISPR screen carried out here is resulting in indels on the target gene, presumably resulting in abolished expression of the target gene product. This is of course problematic for identifying essential genes such as SEC61A1 and this can be circumvented by use of CRISPRi screens. It is not immediately clear to me from the text whether this as a CRISPR KO or a CRISPRi screen. This should be clarified and if a KO screen, then it should be discussed how the essential SEC61A1 came up as a hit.

4) Page. 19, Fig. 3. The followup screen clearly shows that cells transduced with sgRNAs against SEC61A1 are highly resistant against mycolactone cytotoxicity. Given the cell-essential nature of Sec61 in all eukaryotic cells, I am slightly confused about this result. Typically, full blockade or downregulation of Sec61 is highly toxic to cells already around 72 hours (see e.g. PMID: 32067014). The authors should discuss this unexpected result in the text. Perhaps the SEC61A1 KO was not complete in these experiments. Does the sgRNA targeting of SEC61A1 and the used mycolactone treatment block new synthesis of secreted and membrane proteins? This could be tested by for example WB analysis of known Sec61 substrate proteins expressed in THP-1 cells.

5) Page 25. “Therefore, we speculate that mycolactone directly binds to SEC61A1, thereby inhibiting modification of signaling peptides and calcium leakage, which results in accumulation of unfolded proteins in the ER. This process in turn causes ER stress”. This sentence needs to be rewritten. First, it is well established that mycolactone directly binds Sec61 (PMID: 27821549, 32692975) and this should be acknowledged in the text. Second, mycolactone inhibits protein translocation into the ER primarily by blocking access of polypeptides into the ER lumen (or ER membrane for integral membrane proteins). Signal peptide cleavage (not signalling peptide) occurs after the signal peptide has passed through the channel.

Minor points:

1) In the abstract, the authors state that SEC61A1 could be a therapeutic target for Buruli ulcer disease, but I am not convinced how Sec61 could be targeted for this purpose. The authors should clarify their rationale on how modulation of Sec61 activity could produce therapeutic benefits against Buruli ulcer disease keeping in mind that global blockade of Sec61 activity is generally cytotoxic to all eukaryotic cells.

2) Page 13. The authors state that significant THO-1 cell death was induced by 30 ng/ml mycolactone at 48 hours. How does this compare to IC50 values reported elsewhere?

3) Page 15. Authors write that cells were mutagenized using lentiviral gRNA delivery. Is mutagenized the correct word here

4) Page 16. “Cells were then treated with mycolactone, and genomic DNA was prepared from the surviving cells”. Please state the concentration used and the duration of compound treatment.

5) Page 19. “We next explored the possibility that SEC61A1 is involved in mycolactone induced apoptosis.”. As mentioned, the role of Sec61 in mycolactone cytotoxicity is well established and this sentence should be reworded maybe to highlight the less studied form of apoptosis that mycolactone induces.

6) Page 22. The authors discuss the role of mycolactone for inducing a ER stress response. Here, a citation to a proteomic study characterizing the specific form of proteotoxic stress induced by mycolactone was characterized (PMID: 29915147).

7) Page 24. Sec61 protein complex is typically written as Sec61, not SEC61.

8) Page 26. Again, please clarify how Sec61 modulation could allow therapeutic targeting of Buruli ulcer disease.

PLOS authors have the option to publish the peer review history of their article (what does this mean?). If published, this will include your full peer review and any attached files.

Reviewer #1: Yes: Sarah O'Keefe

Reviewer #2: No

Reviewer #3: No
---

## [Decision Letter · Decision Letter 1]

12 Jun 2022

Dear Dr. Suzuki,

Thank you very much for submitting your manuscript "Genome-wide screening identified SEC61A1 as an essential factor for mycolactone-dependent apoptosis in human premonocytic THP-1 cells" for consideration at PLOS Neglected Tropical Diseases. As with all papers reviewed by the journal, your manuscript was reviewed by members of the editorial board and by several independent reviewers. The reviewers appreciated the attention to an important topic. Based on the reviews, we are likely to accept this manuscript for publication, providing that you modify the manuscript according to the review recommendations. 

Please ensure you instigate the editorial changes requested by reviewer 2. In particular, the final sentence should be altered... unfortunately other Sec61 inhibitors replicate (rather than ameliorate) mycolactone action, although it is agreed that modulating the downstream effects of this may be beneficial.

Sincerely,

Rachel E Simmonds, Ph.D.

Guest Editor

Gerd Pluschke

Deputy Editor

Please ensure you instigate the editorial changes requested by reviewer 2. In particular, the final sentence should be altered... unfortunately other Sec61 inhibitors replicate (rather than ameliorate) mycolactone action, although it is agreed that modulating the downstream effects of this may be beneficial.

Reviewer's Responses to Questions

**Key Review Criteria Required for Acceptance?**

**Methods**

-Are the objectives of the study clearly articulated with a clear testable hypothesis stated?

-Is the study design appropriate to address the stated objectives?

-Is the population clearly described and appropriate for the hypothesis being tested?

-Is the sample size sufficient to ensure adequate power to address the hypothesis being tested?

-Were correct statistical analysis used to support conclusions?

-Are there concerns about ethical or regulatory requirements being met?

Reviewer #2: (No Response)

Reviewer #3: The methods for addressing the questions raised by the reviewers are appropriate.

**Results**

-Does the analysis presented match the analysis plan?

-Are the results clearly and completely presented?

-Are the figures (Tables, Images) of sufficient quality for clarity?

Reviewer #2: (No Response)

Reviewer #3: The results are sufficiently presented with the exception of the Western blot images in Figures 1,4 and 5. It is difficult to assess the quality of the Western blotting data as the images are cropped very close to the bands. I would recommend including uncropped WB images in the Supplementary Data section.

**Conclusions**

-Are the conclusions supported by the data presented?

-Are the limitations of analysis clearly described?

-Do the authors discuss how these data can be helpful to advance our understanding of the topic under study?

-Is public health relevance addressed?

Reviewer #2: (No Response)

Reviewer #3: Conclusions are sufficiently supported by the data and study limitations are also discussed suitably.

**Editorial and Data Presentation Modifications?**

Reviewer #2: Minor Points

1. Line 447. "Thasigargin also induced IRE1alpha and EIF2alpha phosphorylation". This sentence is ambiguous as it could be interpreted as indicating phosphorylation of IRE1 rather than an increased protein expression. Please rewrite making the meaning clear.

2. Line 491 . " The Sec61 complex also functions as a channel that mediates calcium leakage from the ER, a process required for stabilization if protein folding". This makes it sound as if it is the calcium leakage that stabilises folding whereas it is the calcium within the ER which is required for protein folding and the loss due to leakage will lead to destabilisation. Please rewrite this sentence to make this clear.

3. Line 508. "..thereby inhibiting the modification of signalling peptides.." It is not the inhibition of signal peptide modification which is causing the mislocalization but the inhibition of translocation into the ER. This sentence needs to be corrected.

4. Line 569. "... SEC61A1 knockout was not completely prevented by mycolactone-induced apoptosis.". I assume you mean mycolactone-induced aopotosis was not completely blocked by SEC61A1 knockout.

5. Line 587. "... to block mycolactone by inhibiting SEC61A..." Inhibition of SEC61A1 will NOT block the actions of mycolactone. Please take this out or rephrase to make it clear that you mean targeting of downstream effects.

Reviewer #3: I would recommend accepting the article in the current form if full Western blotting images are provided.

**Summary and General Comments**

Reviewer #2: The authors have adequately addressed most of my concerns. I think the paper should be accepted if the above editorial amendments are made

Reviewer #3: The authors have addressed all of the reviewer suggestions by new experiments or modifications to the text. I recommend publication of the article in the current form.

PLOS authors have the option to publish the peer review history of their article (what does this mean?). If published, this will include your full peer review and any attached files.

Reviewer #2: No

Reviewer #3: No

Figure Files:

Data Requirements:

Reproducibility:

References

---

## [Editor Report · Decision Letter 2]

18 Jul 2022

Dear Dr. Suzuki,

We are pleased to inform you that your manuscript 'Genome-wide screening identified SEC61A1 as an essential factor for mycolactone-dependent apoptosis in human premonocytic THP-1 cells' has been provisionally accepted for publication in PLOS Neglected Tropical Diseases.

Best regards,

Rachel E Simmonds, Ph.D.

Guest Editor

Gerd Pluschke

Section Editor

---

## [Editor Report · Acceptance letter]

4 Aug 2022

Dear Dr. Suzuki,

We are delighted to inform you that your manuscript, "Genome-wide screening identified SEC61A1 as an essential factor for mycolactone-dependent apoptosis in human premonocytic THP-1 cells," has been formally accepted for publication in PLOS Neglected Tropical Diseases.

Best regards,

Shaden Kamhawi

co-Editor-in-Chief

Paul Brindley

co-Editor-in-Chief
